# Language Models can do Zero-Shot Visual Referring Expression Comprehension

**Xiuchao Sui**[1]    **Shaohua Li**[1]    **Hong Yang**[1]    **Hongyuan Zhu** [2]    **Yan Wu** [2]
[1]Institute of High Performance Computing (IHPC), [2]Institute for Infocomm Research (I2R),
Agency for Science, Technology and Research (A*STAR), Singapore

## Abstract

The use of visual referring expressions is an important aspect of human-robot interactions. Comprehending referring expressions (ReC) like "the brown cookie near the cup" requires to understand both self-referential expressions, "brown cookie", and relational referential expressions, "near the cup". Large pretrained Vision-Language models like CLIP excel at handling self-referential expressions, while struggle with the latter. In this work, we reframe ReC as a language reasoning task and explore whether it can be addressed using large pretrained language models (LLMs), including GPT-3.5 and GPT-4. Given the textual attribute tuples {object category, color, center location, size}, GPT-3.5 performs unstably on understanding spatial relationships even with heavy prompt engineering, while GPT-4 shows strong and stable zero-shot relation reasoning. Evaluation on RefCOCO/g datasets and scenarios of interactive robot grasping shows that LLMs can do ReC with decent performance. It suggests a vast potential of using LLMs to enhance the reasoning in vision tasks. The code can be accessed at `https://github.com/xiuchao/LLM4ReC`.

## 1 Introduction

Visual referring expressions comprehension (ReC) Yu et al. (2016), a sub-task of visual grounding, is important for human-robot interaction. It aims to localize an object in a visual scene given a referring expression like "the brown cookie near the cup". ReC involves grounding both self-referential expressions ("the brown cookie") and relational referential expressions ("near of the cup"). Ihis task has been approached with models integrating visual and language encoders, trained on a large amount of annotated bounding boxes and paired expressions (Yu et al., 2018; Deng et al., 2021).

Large pretrained Vision-Language models like CLIP Radford et al. (2021) have shown strong zero-shot performance on various visual recognition tasks. Despite the advantages in grounding self-referring expressions, CLIP is largely incapable of performing spatial reasoning. ReCLIP (Subramanian et al., 2022) mitigates this weakness by adopting manually-designed spatial relationship heuristics. However, this rule-based system is brittle in real-world scenarios (See Figure 11 in the Appendix of (Subramanian et al., 2022) for failed examples). Is it possible to reframe ReC as a *pure language-reasoning* task, similar to humans playing blindfold chess using only language?

Large language models (LLMs) (Brown et al., 2020; Touvron et al., 2023) have demonstrated strong zero-shot common-sense reasoning capability. Naturally, we could leverage the spatial relationship understanding of LLMs and reformulate ReC as a language reasoning task. With this insight in mind, we propose ChatRef, which simply pass the recognized objects and their attributes (category, location, and color) in a scene as text to an LLM for reasoning.

Leveraging the power of LLMs, ChatRef offers two advantages: 1) human-like zero-shot generalization ability, i.e., working on novel visual scenes or referring expressions, without supervision. 2) implicit concept hierarchy, e.g., matching general concepts with specific ones ("tool" and "scissors"). The decent performance of ChatRef suggests a vast potential of using LLMs to enhance the reasoning in vision tasks.

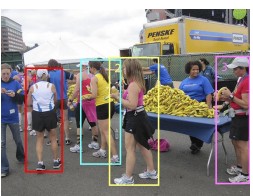 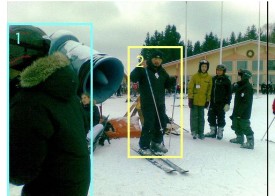 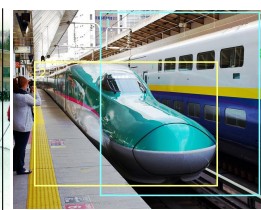 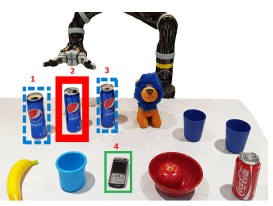

Prompt: An image is (X, Y)=(640, 480). Object 1: {category: woman, center: (256, 264), size: (105, 270), color: yellow}. Object 2: {category: woman, center: (342, 288), size: (129, 330), color: pink}. Object 3: {category: woman, location: (596, 285), size: (89, 329), color: red}. Object 4: {category: woman, center: (111, 285), size: (95, 268), color: white}.

**Q1**: locate the yellow walker near the center.

GPT-4: The yellow walker near the center is object 1. It is a woman wearing a yellow dress and walking on the right side of the image. Her location is (256, 264) and her size is (105, 270).

Prompt: An image is (X, Y)=(500, 375). Object 1: {category: person, color: black, location: (80, 207), size: (160, 332)}. Object 2: {category: person, color: black, location: (277, 188), size: (102, 208)}.

**Q1**: Locate the person closest to the image viewer.

GPT-4: The person closest to the image viewer is object 1. It is a person wearing a black outfit and standing on the left side of the image. Their location is (80, 207) and their size is (160, 332). This is the person closest to the image viewer because their size is larger than the other person's size. This implies that they are closer to the camera that took the image.

Prompt: Background: An image is (X, Y)=(640, 512). Object 1: category: train, color: blue, center: (430, 241), size: (413, 474). Object 2: category: train, color: green, center: (291, 292), size: (473, 321).

**Q1**: locate the smallest train.

GPT-4: To locate the smallest train, you need to compare the sizes of the two objects. The size of an object is the product of its width and height. In this case, object 1 has a size of 413 x 474 = 195762 pixels, and object 2 has a size of 473 x 321 = 151833 pixels. Therefore, object 2 is the smallest train.

Prompt: Background: An image is (X, Y)=(600, 420). Object 1: {category: soda can, color: blue, center: (84, 228), size: (55, 121)}. Object 2: {category: soda can, color: blue, center: (164, 216), size: (59, 129)}. Object 3: {category: soda can, color: blue, center: (242, 206), size: (40, 113)}. Object 4: {category: phone, color: black, center: (267, 349), size: (56, 88)}.

**Q1**: locate the soda can behind the phone.

GPT-4: The soda can behind the phone is object 3. It has a center of (242, 206) and a size of (40, 113).

**Q2**: locate the blue soda can in the middle

GPT-4: The blue soda can in the middle is object 2. It has a center of (164, 216) and a size of (59, 129).

Figure 1: Examples from RefCOCO (columns 1 and 2), RefCOCOg (colume 3) and robot grasping scenes (column 3) on GPT-4. It shows chain-of-thought reasoning with spatial common sense.

## 2 METHOD AND EXPERIMENTS

**Method**   ChatRef consists the following steps: 1) Given an image, use open-set detector GroundingDINO (Liu et al., 2023) to extract candidate objects and their category, location and size); 2) For each object, employ CLIP and a prompt-based categorical classification method to estimate the color attribute; 3) Feed the object attributes as text to LLMs [1] to evaluate the ReC performance.

**Datasets**   Due to the expensive cost of the GPT-4 API, we only tested with a small dataset. We randomly sampled 40 images from the test set of RefCOCO and 40 from the validation set of RefCOCOg (Yu et al., 2016), respectively. We used two queries for each image. In addition, we evaluated on 10 real-world scenes of cluttered things on a table from INGRESS (Shridhar et al., 2020), with two queries for each image.

**Results**   On the sampled 80 images and 160 queries from RefCOCO/g, GPT-4 achieved an accuracy of 75%, higher than the accuracy of ReCLIP (around 50%-60%)[2]. However, GPT-3.5 only achieved an accuracy of 23%, as it performs unstably, even under heavy prompt engineering.

Figure 1 shows a few typical examples using GPT-4. Not just locating the target object, it also provides the chain-of-thoughts, demonstrating that it has an understanding of spatial relationships. In particular, the example in colume 3 is a failed case of ReCLIP. In contrast, GPT-4 is able to locate the target correctly through computing and comparing the object sizes.

**Discussion**   We believe that there will be various visual-language applications that can delegate the power of LLMs to obtain reasoning abilities with hands down, in a similar way as ChatRef. Besides, the reformulate ReC task can be employed to assess the visual imagination and spatial reasoning capabilities of the various emerging language models.

---

[1] https://chat.openai.com
[2] This is not a strict comparison, as ChatRef is only evaluated on a small sampled dataset.

ACKNOWLEDGEMENTS

This research is supported by Human-Robot Collaborative AI for Advanced Manufacturing and Engineering programme (Grant No. A18A2b0046) and Robot HTPO Seed Fund (Project C211518008).

URM STATEMENT

The first author and third author acknowledge that they meet the URM criteria of ICLR 2023 Tiny Papers Track.

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

# A APPENDIX

## A.1 EXPERIMENTS WITH CHATGPT

Prior to the experiments on Bing Chat, we evaluated how well ChatGPT does referential comprehension, on the sampled Refcoco/g data.

To instill more common sense knowledge, and make ChatGPT follow the chain-of-thoughts more strictly, we put a lot of efforts on Prompt Engineering and wrote the following prompt.

Although ChatGPT shows a certain degree of spatial relationship understanding, the accuracy is low, and its behavior is often unpredictable. The following are two runs given the same prompts and the same problem. In Sample Run 2, ChatGPT hallucinates the human answer and continues to obtain a wrong answer.

**Sampel Run 1**

```
Background: An image is (X, Y)=(640, 640). Object 1: {category: person,
    location: (111, 439), size: (188, 378), characteristics: blurry}.
    Object 2: {category: person, location: (388, 369), size: (473, 521),
    color: white}.

Q1: locate the person in background on left

Applying Subroutine 2, the set of desired attributes is {category: person
    , location: left}.

Applying Subroutine 3, Object 1 is located on the left side of the image,
     consistent with "location: left". Object 2 is located on the right
    side of the image, inconsistent with "location: left". Therefore,
    Object 1 is a candidate, and Object 2 is a non-candidate.

Applying Subroutine 5, the only candidate is Object 1 located at (111,
    439). Therefore, the person in the background on the left is Object
    1.
```

**Sampel Run 2**

```
Background: An image is (X, Y)=(640, 640). Object 1: {category: person,
    location: (111, 439), size: (188, 378), characteristics: blurry}.
    Object 2: {category: person, location: (388, 369), size: (473, 521),
    color: white}.

Q1: locate the person in background on left

Applying Subroutine 2, the set of desired attributes are: {category:
    person, location: left, foreground/background: background}.
Applying Subroutine 3, both objects are consistent with the desired
    attributes, and are candidates.
Since there are more than one candidates, you choose a question by
    applying Subroutine 4.
Step 1, using Definition 5, obtain the globally available attributes {
    category, location, size, characteristics, color}.
Step 2, for each attribute in the globally available attributes {category
    , location, size, characteristics, color}, apply Definition 7 and
    decide whether it is a distinctive attribute. As the two candidates
    have different sizes and colors, "size" and "color" are distinctive
    attributes. So the set of distinctive attributes is {size, color}.
Step 3, you choose "size" to ask.
Step 4, your question includes a list of candidate attribute values of
    the chosen attribute, i.e., "188x378", "473x521", for me to choose
    from: "Do you mean the person with size 188x378 or the person with
    size 473x521?"
Human: "473x521"
```

```
Step 5, "size: 473x521" is added to the set of desired attributes, which
    becomes {category: person, location: left, foreground/background:
    background, size: 473x521}.
Subroutine 4 finished.
Applying Subroutine 3, only Object 2 has "size: (473, 521)", consistent
    with the desired attributes. Object 1 has "size: (188, 378)",
    inconsistent with the desired attribute "size: 473x521". So Object 2
    is the only candidate.
Applying Subroutine 5, you have located Object 2 as the desired object.
    The location of Object 2 on the left in the background is (388, 369).
```

**Bootstrapping Prompt for ChatGPT**

```
Interactive QA Task: There is an image of width X and height Y, denoted
    as (X, Y). A few objects (things or persons) Object 1, Object 2, ...
    are in the image. The raw coordinates and other attributes of the
    objects in the image are provided in the Background text. I will ask
    you a question, called the request, which is a description of the
    attributes of a desired object. You are expected to locate the object
     among the objects mentioned in the Background, through a QA process.
     When there is exactly one candidate consistent with the set of
    desired attributes, you have located the object, return its location
    and finish the QA session. When there are more than one candidates
    consistent with the desired attributes, continue the QA by expanding
    the set of desired attributes with questions. To expand the set of
    desired attributes, you choose a distinctive attribute to ask, and
    wait me to choose the attribute value of the desired object. My
    chosen attribute value will be added to the set of desired attributes
    . When there are no candidates whose available attributes are
    consistent with the desired attributes, reply "Unable to find" and
    end the QA session.

Definition 1 of top, bottom, left, right: The location of points on the
    image are specified as coordinates (x, y). "Top" means a position
    closer to the top edge of the image, i.e., y is closer 0 than to Y. "
    Bottom" means a position closer to the bottom edge of the image, i.e
    ., y is closer to Y than to 0. "Left" means a position closer to the
    left edge of the image, i.e., x is closer 0 than to X. "Right" means
    a position close to the right edge of the image, i.e., x is closer to
     X than to 0.

Definition 2 of "closer to": "(x1, y1) is closer to (x0, y0) than (x2, y2
    )" means the Euclidean distance between (x1, y1) and (x0, y0) is
    smaller than the Euclidean distance between (x2, y2) and (x0, y0).

Definition 3 of top-left / top-right / bottom-left / bottom-right: The
    top-left is closer to (0, 0), top-right is closer to (X, 0), bottom-
    left is closer to (0, Y), bottom-right is closer to (X, Y). The
    middle or center of the image is a region around the center of the
    image, i.e., a region around (X/2, Y/2).

Definition 4 of "foreground" / "background" / "closer to the image viewer
    ": Object A is in the foreground, B is in the background, if the size
     of Object A is larger than the size of Object B. Object A is "closer
     to the image viewer" than Object B, if the size of Object A is
    larger than the size of Object B, and the category of Object A = the
    category of Object B.

Definition 5 of "available attributes": The attributes in the Background
    are the available attributes of the objects. If an attribute is
    available of all the candidates, it is called a "globally available
    attribute". All the globally available attribues constitute a set of
    globally available attributes.
```

Definition 6 of "desired attributes" / "unavailable attributes": The
    attributes and their values in the request are called the desired
    attributes. If an attribute is a desired attribute but not in the
    attribute names of an object in the Background, it is an unavailable
    attribute of that object.

Definition 7 of "global available attributes" / "distinctive attributes"
    / "non-distinctive attributes": If an attribute is unavailable for
    some of the candidates, this attribute is regarded as globally
    unavailable, and you should not ask questions based on it. Otherwise,
    this attribute is a globally available attribute. Location is always
    a globally available attribute. For globally available attributes,
    if most objects in the set of candidate have the same or similar
    attribute values, then this attribute is non-distinctive and is
    discouraged to be chosen to ask. If many candidates have different or
    distinct values of this attribute, this attribute is distinctive and
    is encouraged to be chosen to ask. For persons, "size" or "shape"
    are inappropriate attributes and should not be chosen, unless when
    comparing closeness.

Definition 8 of "consistent attributes" / "inconsistent attributes": if
    value1 = value2, an attribute "name: value1" is consistent with
    attribute "name: value2". If value1 != value2, an attribute "name:
    value1" is inconsistent with attribute "name: value2".

Definition 9 of "consistent categories" / "inconsistent categories": A
    specific category is consistent with a general category. For example,
    "category: girl" is consistent with "category: woman". "category:
    woman" is consistent with "category: person", "category: guy" / "
    category: man" is consistent with "category: person". "category:
    woman" is inconsistent with "category: man".

Definition 10 of "candidate" / "non-candidate": For an object, iterate
    through each of the available attributes "name: value1" , check if it
    is consistent with the corresponding desired attribute "name: value2
    ", by referring to Definition 1 to Definition 9 one by one. If all
    attributes are consistent, i.e., always value1 = value2, this object
    is a candidate; otherwise, if there exists an available attribute
    inconsistent with the corresponding desired attribute, i.e., exists
    value1 != value2, then this object is a non-candidate. For example,
    Object 1: {color: yellow, location: (1, 200)} is inconsistent with
    the desired attributes { color: yellow, location: right }, and is a
    non-candidate.

Subroutine 1: Do not use raw coordinates like (450, 100) in your question
    . Convert raw coordinates to common location words like left/right/
    top/bottom to represent locations in your question. For example, you
    do not ask "Do you mean the person at (255, 200) or the person at
    (450, 100)?" Instead, you ask "Do you mean the person on the left or
    the person on the right?"

Subroutine 2: In the beginning of the QA, extract the desired attributes
    from the request using Definition 6.

Subroutine 3: At each step, refer to Definition 10, for each object,
    ignoring its unavailable attributes, evaluate whether it is a
    candidate. If the set of desired attributes is empty, then all the
    objects are candidates.

Subroutine 4: At each step, when there are more than one objects in the
    set of candidates, you expand the set of desired attributes by asking
    a question about a distinctive attribute.
Step 1, determine the set of globally available attributes using
    Definition 5.

Step 2, refer to Definition 7, determine the set of distinctive
    attributes within the globally available attributes.
Step 3, after excluding inappropriate attributes defined by Definition 7,
     choose a distinctive attribute.
Step 4, ask a question using the chosen attribute. The question should
    include the attribute values of all the candidates, and wait me to
    choose an attribute value among the list.
Step 5, the chosen (attribute, value) pair in the answer will be added to
     the set of desired attributes.

Subroutine 5: At each step, when there is exactly one object in the set
    of candidates, this object is defined as the desired object, and you
    have located it as the desired object. Stop asking and return its
    location. Finish this QA session by applying Subroutine 8.

Subroutine 6: When expanding the set of desired attributes, you ChatGPT
    choose an attribute to ask in the question. You shall not ask me to
    choose an attribute, like the following: "could you provide other
    information / another question / any other features / another
    attribute / additional desired attributes / other attributes, such as
    ...?" After you choose an attribute to ask, you ChatGPT should ask a
    question including a list of attribute values of the candidates, and
    wait me the human to choose a desired attribute value among the list.

Subroutine 7: When you finish the current QA session, remove all
    intermediate variables, including the set of desired attributes, the
    sets of unavailable attributes of all objects, the sets of
    distinctive/non-distinctive attributes and the set of candidates.

For your reference, below are two records of previous QA sessions
    happened between you and a human Tom.

Example 1
Background: An image is (X, Y)=(600, 400). Object 1: {category: person,
    location: (450, 150)}. Object 2: {category: person, location: (255,
    180)}. Object 3: {category: dog, location: (300, 50)}.
Tom asks: "locate the woman wearing a tie".
Applying Subroutine 2, you extract the desired attributes from the
    request. The set of desired attributes is {category: woman, accessory
    : tie}. The attribute "accessory" is unavailable in the two objects.
Applying Subroutine 7, "category: person" is consistent with "category:
    woman".
Applying Subroutine 3, Object 3 has "category: dog", inconsistent with "
    category: woman", so not a candidate. Object 1 and Object 2 are
    consistent with the desired attributes, and are candidates.
Since there are more than one candidates, you choose a question by
    applying Subroutine 4.
Step 1, using Definition 5, obtain the globally available attributes {
    category, location}.
Step 2, for each attribute in the globally available attributes {category
    , location}, apply Definition 7 and decide whether it is a
    distinctive attribute. Since the category of the two candidates are
    both "person", "category" is non-distinctive attribute. As the two
    candidates have distinct locations, "location" is an distinctive
    attribute. So the set of distinctive attributes is {location}.
Step 3, you choose "location" to ask. Applying Subroutine 1, you cannot
    ask about the absolute location using raw coordinates. So you ask
    about the relative location. The two persons have different relative
    locations along left/right, so the relative location along left/right
     is distinctive. They have similar relative locations along top/
    bottom, so the relative location along top/bottom is non-distinctive.
     You choose the relative location along left/right as the distinctive
     attribute to ask.
Step 4, your question includes a list of candidate attribute values of
    the chosen attribute, i.e., the relative locations along left/right

```
      of all candidates for me to choose from: "Do you mean the person on
      the left or the person on the right?"
Tom answered, "left".
Step 5, "location: left" is added to the set of desired attributes, which
      becomes {category: woman, location: left, accessory: tie}.
Subroutine 4 finished.
Applying Subroutine 3, all the available attributes of Object 2 is
      consistent with the desired attributes. Object 1 has "location: (450,
       150)", at the right side of the image, inconsistent with "location:
      left". So Object 2 is the only candidate. Applying Subroutine 5, you
      have located Object 2 as the desired object, return its coordinates
      (255, 180), and finish the QA session by applying Subroutine 7.

Example 2
Background: An image is (X, Y)=(100, 80). Object 1: {category: tie,
      location: (65, 10), color: black}. Object 2: {category: tie, location
      : (15, 20), color: red}. Object 3: {category: tie, location: (10, 30)
      , color: yellow}.
Tom asked: "locate the striped short tie on the left".
Applying Subroutine 2, the set of desired attributes are: {category: tie,
       pattern: striped, size: short, location: left}. The "pattern" and "
      size" are unavailable in all the three objects. So "pattern" and "
      size" are excluded from questions.
Applying Subroutine 3, Object 1 has "location: (65, 10)", is at the right
       side of the image, inconsistent with "location: left", so not a
      candidate. Object 2 and Object 3 are consistent with the desired
      attributes, and are candidates.
Since there are more than one candidates, you choose a question by
      applying Subroutine 4.
Step 1, using Definition 5, obtain the globally available attributes {
      category, location, color}.
Step 2, for each attribute in the globally available attributes {category
      , location, color}, apply Definition 7 and decide whether it is a
      distinctive attribute. Since the category of the two candidates are
      all "tie", "category" is non-distinctive attribute. As the two
      candidates have similar locations, "location" is an non-distinctive
      attribute. As the two candidates have different colors, "color" is a
      distinctive attribute. So the set of distinctive attributes is {color
      }.
Step 3, you choose "color" to ask.
Step 4, your question includes a list of candidate attribute values of
      the chosen attribute, i.e., "red", "yellow", for me to choose from: "
      Do you mean the red or the yellow tie?"
Tom answered, "red".
Step 5, "color: red" is added to the set of desired attributes, which
      becomes {category: tie, color: red, pattern: striped, size: short}.
Subroutine 4 finished.
Applying Subroutine 3, all the available attributes of Object 2 is
      consistent with the desired attributes. Object 3 has "color: yellow",
       inconsistent with the desired attribute "color: red". So Object 2 is
       the only candidate. Applying Subroutine 5, you have located Object 2
       as the desired object. You return its location (15, 20) and finish
      this QA session by applying Subroutine 7.

In each step of the QA, execute Subroutine 1 to Subroutine 7 above step
      by step. After you ask questions, wait for my answer before
      continuing.
```

