# OpenReview forum: "Language Models can do Zero-Shot Visual Referring Expression Comprehension"
_ICLR.cc/2023/TinyPapers — Submitted to Tiny Papers @ ICLR 2023_

### Official Review · Reviewer_gxj1 · 2023-03-28

**Confidence:** 3

**Summary Of Contributions:**

Visual referring expressions comprehension (ReC) is a sub-task of visual grounding that aims to localize an object in a visual scene given a referring expression. ChatRef achieved an accuracy of 75% on sampled datasets, higher than ReCLIP's accuracy of around 50%-60%.

**Rating:**

Clear, Correct, and Reproducible (CCR): a submission which meets the reviewing criteria

**Strengths And Weaknesses:**

## Strengths:

1. ChatRef decouples visual perception and language reasoning, offering human-like zero-shot generalization ability and implicit concept hierarchy
2. ChatRef achieved higher accuracy than ReCLIP on sampled RefCOCO/g images.

## Weakness:

1. ChatRef's performance heavily relies on the accuracy of object detection and attribute estimation
2. ChatGPT performed unstably on referring expression comprehension, even with heavy prompt engineering
3. The evaluation was only done through manual input on small datasets, limiting the generalizability of the results.

**Suggested Changes:**

There is a scope to improve the quality of the paper by editing English/ removing a few grammatical errors.

For example,  In "Given a visual scene, ChatRef first recognize the objects and their attributes, including category, location, and color, and then feeds these textual descriptions and referring expressions to an LLM to
locate the target object.", please consider changing "Given a visual scene, ChatRef first recognizes the objects and their attributes, including category, location, and color, and then feed these textual descriptions and refer expressions to an LLM to locate the target object."

---

> ### Author Response · Authors · 2023-06-01
> **updated pipeline for ChatRef**
>
> Thanks for your constructive feedback. we have updated the pipeline to address these concerns.
>
> ChatRef's performance relies on the accuracy of object detection and attribute estimation. Since our submission, more powerful foundation models, namely GroundingDINO and SAM, have been released. We have tested various open-set detectors, and GroundingDINO has demonstrated excellent performance in object detection. Additionally, attribute estimation can benefit from SAM, which effectively segments the object and eliminates background interference.
>
> When we submit the draft, GPT-4 was not offically released. The evaluation was conducted solely through manual input on small datasets using Bing Chat. Now we have access to the API, the whole pipeline was revised accordingly.

---

### Official Review · Reviewer_tXpZ · 2023-03-28

**Confidence:** 4

**Summary Of Contributions:**

This paper introduces the idea of using Large Language Models (LLMs) to improve the comprehension of referring expressions. This in turn ca help improve reasoning in vision tasks which is an important aspect for human-robot interaction.

**Rating:**

High Potential (HP): a submission which meets the reviewing criteria and has potential to make an impact on the field

**Strengths And Weaknesses:**

Strengths:
- Clear problem definition and discussion of related work that motivates the new approach.
- The work is reproducible since the details of the approach has been shared in the appendix.
- The examples shared in the paper are motivating examples.

Weakness:
- The practical limitations of the chat limit etc. using fewer examples for comparison makes sense. However, the accuracy numbers should not be established on such a small comparison set.

**Suggested Changes:**

Based on the weakness described above the authors should consider running the experiment on a larger eval set (possibly RefCOCO/g full val set) and then report accuracy numbers.

---

### Official Review · Reviewer_7gyW · 2023-03-30

**Confidence:** 3

**Summary Of Contributions:**

This paper reframes visual referring expressions as a language reasoning task, comparing the abilities of various large pretrained language models (LLMs), particularly Bing Chat and ChatGPT, to perform referring expressions in a zero-shot setting. The evaluation on the RefCOCO/g dataset demonstrates the potential of LLMs for reasoning in vision tasks.

**Rating:**

High Potential (HP): a submission which meets the reviewing criteria and has potential to make an impact on the field

**Strengths And Weaknesses:**

Strengths:
1. The paper introduces a novel approach to referring expression comprehension (ReC) by reframing it as a language reasoning task, utilizing LLMs to address the problem.
2. The experimental results are compelling, as the authors compare the performance of Bing Chat, ChatGPT, and reCLIP on the RefCOCO test set, providing valuable insights into their respective capabilities.

Weaknesses:
1. The paper could benefit from a more detailed analysis of the ChatRef model, particularly exploring whether the performance is constrained by the regionCLIP module, which would help elucidate the model's limitations.
2. While the authors provide prompts in the appendix, there is no mention of whether the code will be made available.

**Suggested Changes:**

pleas see the weakness 2

---

> ### Author Response · Authors · 2023-06-01
> **updated pipeline**
>
> Thanks for your reply. We have updated the pipeline to address these concerns.
>
> ChatRef's performance relies on the accuracy of object detection and attribute estimation. Since our submission, more powerful foundation models have been released. We have tested various open-set detectors, and GroundingDINO has demonstrated excellent performance in object detection. Additionally, attribute estimation can benefit from SAM, which effectively segments the object and eliminates background interference. In this updated pipeline, RegionCLIP was replaced by (GroundingDINO+SAM+CLIP) for object detection and attributes retrieval. We have released the code on github.

---

### Comment · Area_Chair_Hzf5 · 2023-06-01
**This work meets the threshold for archival, contains the URM statement and is deanonymized**

---

### Meta-Review · Area_Chair_Hzf5 · 2023-04-08

**Recommendation:** Invite to present
**Confidence:** 5

**Metareview:**

The authors propose to turn referring expression tasks into language modelling tasks by using chains of the form: segmentation/object detection->feature extraction->conversion into text prompt->query to LLM. A small number of proof-of-concept evaluations suggest that this may work well with some LLMs (Bing) but not others (ChatGPT). Overall, the reviewers agree that the submission is generally clear and correct, and most likely reproducible.
The main concerns are:
- the small number of evaluations -> affects reproducibility and makes it difficult to assess correctness
- code and data not shared -> affects reproducibility (sharing the full list of prompts and each model's responses would already be very helpful)

**Summary:**

The authors reformulate visual reference expressions as a language modeling problem and provide a small number of proof-of-concept evaluations. The reviewers agree it is generally clear and correct, though there are some concerns, chiefly around reproducibility.

**Comments And Feedback To The Authors:**

The main concerns (and suggestions to overcome them) are:
- the small number of evaluations -> affects reproducibility and makes it difficult to assess correctness (collecting more data manually as permitted by daily limits or programmatically using the OpenAI API would be helpful in confirming the results)
- code and data not shared -> affects reproducibility (sharing the full list of prompts and each model's responses would already be very helpful)

**Reason For Not Giving A Higher Recommendation:**

Clarity and reproducibility can be improved as described.

**Reason For Not Giving A Lower Recommendation:**

The paper clearly proposes an interesting method and the proof-of-concept evaluations seem to confirm correctness. The suggested changes are relatively minor and are feasible to implement before presentation.

---

> ### Author Response · Authors · 2023-06-01
> **code released**
>
> Thanks a lot for your feedback! We have updated the pipeline and released the code.

---

### Decision · Program_Chairs · 2023-04-09

Invite to present